# OpenReview forum: "PopAlign: Population-Level Alignment for Fair Text-to-Image Generation"
_ICLR.cc/2025/Conference — ICLR 2025 Conference Withdrawn Submission_

### Official Review · Reviewer_d3aE · 2024-11-04

**Soundness:** 3
**Presentation:** 2
**Contribution:** 2
**Rating:** 5
**Confidence:** 3

**Summary:**

The paper introduces PopAlign, an approach to align Text-to-image models towards population-level preferences. This is done by extending the direct preference optimization objective that works on pairs of images, to sets of images that represents population preferences. Experimental results show that the method is able to generate less biased (in terms of racial and gender attributes) images while maintaining high image quality.

**Strengths:**

- The paper was well written and easy to follow.
- The method is clear and easy to use and does not degrade the quality of the generations.

**Weaknesses:**

- More details are needed on how well the method generalizes to new prompts. Sec 6.3 mentions that 100 prompts are manually written for evaluation, how similar are these prompts to those used for training? Does it generalize to actions, or the more general prompts from LAION Aesthetics dataset, style and personal descriptors used by Shen et al., or to multiple people? Sec 6.5 evaluates on generic prompts but not for fairness.
- Scalability of the method. The categorial attributes of gender and race were used in the paper, it would have been useful to demonstrate the method with other types of attributes.
    - A classifier and face detector are used to filter out inconsistent or ambiguous images. There may be other attributes where such models need to be trained separately. An ablation study of the importance of this step would also be useful.
    - It is not clear how prompt augmentation is done, e.g., manually or with another language model, and how this it should be done for continuous attributes e.g., age.
- The method seems to be computationally expensive, compared to the baselines as it requires generating additional data and finetuning. An additional column specifying the costs of the method and baselines would be useful in understanding the tradeoffs.

**Questions:**

- What are some of the challenges in scaling the method to more attributes or non binary attributes (e.g., age, culture norms) and to more complicated generations where there are multiple entities? Is the bottleneck mostly in the generation (compositional generation, sanity check) step?
- How sensitive is the method to the filtering stage described in  Sec. 4.1? E.g., what would be the performance of the method without the sanity checks?
- What is the computation cost of the entire pipeline for the methods in Tab. 1?
- How many sets of images were evaluated in Fig. 5’s study and how many users were involved?

---

> ### Author Response · Authors · 2024-11-21
>
> **W1: Generalization to New Prompts**
>
> We have addressed this by:
> 1. Including experiments on **LAION-Aesthetic** and **Personal Descriptors** datasets as suggested in Table 2, demonstrating zero-shot generalization to unseen prompts.
> 2. Extending evaluations to complex setups, such as **multi-person generations** for sexuality biases Table 2 and Appendix Figure 10.
> 3. Providing qualitative results on diverse use cases in Appendix Figure 10 and listing all evaluation prompts in Appendix L.
>
> **W2.1: Demonstrating Additional Attributes**
>
> We include results for **age** and **sexuality** biases in Table 2, showing that PopAlign effectively addresses diverse and challenging attributes (Section 5.1, L298).
>
> **W2.2: Importance of Filtering and Ablation Studies**
>
> In general, the face detector and classifier is an optional part of PopAlign, unlike previous methods which required running classifiers online during the training process. For new experiments (table 2) on age and sexuality biases, we skipped the classifier part, results show that PopAlign can still mitigate these biases considerably.
>
> Most importantly, the classifier demonstrates a high recall. In Appendix B1 Table 5, the classifier has an overall recall of 99.8 for vanilla SDXL. Which means that most of the SDXL generation successfully adhere to the identity specification provided in the augmented prompt.
>
> **W2.3: Prompt Augmentation Details**
>
> Specifically, we use text keywords such as "male", "Asian", “gay”, “lesbian”, “young”, “old” to directly augment the prompt by inserting them to the prompt. Our method works as long as the model can recognize these keywords and generate images accordingly. Unfortunately, we cannot directly prompt a specific numerical age (e.g. 24 years old). But our results in table 2 show that  even with simple keywords, Popalign can achieve decent performance on a wide range of tasks.
>
>
> **W3: Computational Cost**
>
> (Appendix B2) We acknowledge that Popalign is more expensive than training-free methods such as CADS and Dynamic-CFG, and generally more expensive than prompt-injection methods such as iti-Gen. However, compare with existing alignment methods such as Diffusion-DPO and fine-tuning methods for fairness such as aDFT, Popalign is not expensive.
>
> Concretely, Diffusion-DPO used **16 A100 GPUs** with a gradient accumulation of 128 steps, and a global batch size of 2048. It trained for 2000 steps, or **roughly a week** (our estimate).
>
> By comparison, PopAlign trains on **4 A5000 GPUs** with no gradient accumulation and a global batch size of 8. It trained for 750 steps, or **8 hours.**
>
> While aDFT freezes the unet and only trains the text-encoder using lora, it requires generating images, running classifiers, and extracting DINO and CLIP features during the training. It takes **48 hours on 8 NVIDIA A100 GPU** (original author) and **3.5 days on our hardware.**  Hence, PopAlign is considerably less expensive than other methods.
>
> **Q1: Challenges in Scaling**
>
> We have successfully demonstrated that our method can work with age and culture norms such as sexuality, and generations involving multiple entities (sexuality). In general, our method can work as long as there is a reasonable way to obtain a sample wining population.
> However, we note that we followed previous work aDFT and convert the continuous attributes “age” to categorical labels “old” and “young”. It is very difficult to obtain a perfect continuous uniform distribution of different ages as we cannot naively prompt the model to generate a person of age X. We recognize that this is a limitation. However, this limitation is shared by prior works, which either cannot address continuous attributes or convert them to discrete ones.
>
> **Q2: Bottleneck in Filtering**
>
> PopAlign is robust even without sanity checks (see W2.2).
>
> **Q3: Computation Cost**
>
> See W3 for detailed comparisons of computational efficiency.
>
> **Q4: Human Evaluation Details**
>
> We provide additional information about human evaluation in Sec 6.3 (L418) and Appendix E. Specifically, we provide instructions given to human annotators in Appendix E. In total, we collected 300 anonymous responses for 100 prompts. For each prompt, we present 5 images generated by each model. The inter-annotator agreement (Krippendorff’s Alpha) is 0.81.

---

> > ### Author Response · Authors · 2024-12-01
> >
> > Dear Reviewer d3aE
> >
> > As we are approaching the end of discussion period, we would greatly appreciate your feedbacks on our rebuttals. We have provided additional experiments and discussions as requested, such as generalization to new datasets and computation cost.
> >
> > Best, Authors.

---

> > > ### Author Response · Authors · 2024-12-02
> > >
> > > Dear Reviewer d3aE
> > >
> > > As we are approaching the **last day** of discussion period, we would greatly appreciate your feedbacks on our rebuttals. We have provided additional experiments and discussions as requested, such as generalization to new datasets and computation cost.
> > >
> > > Best, Authors.

---

> > > > ### Author Response · Authors · 2024-12-03
> > > >
> > > > Dear Reviewer d3aE
> > > >
> > > > Today is the **last day** of discussion period, we would greatly appreciate your feedbacks on our rebuttals. We have provided additional experiments and discussions as requested, such as generalization to new datasets and computation cost.
> > > >
> > > > Best, Authors.

---

### Official Review · Reviewer_d8f1 · 2024-11-04

**Soundness:** 2
**Presentation:** 3
**Contribution:** 3
**Rating:** 5
**Confidence:** 4

**Summary:**

This work aims at tackling biases of text-to-image (T2I) models that appear at the population. As shown by multiple previous work, T2I models tend to over-represent certain parts of the learned distribution when prompted with underspecified prompts. This work focuses on the societal implications this phenomena might have, aiming at tackling gender and ethnicity biases. To do so, the authors proposed PopAlign, an alignment strategy for pre-trained T2I models that goes beyond pairwise comparisons as common approaches for incorporating human preferences such as RLHF and DPO. Empirical evaluation was carried out considering the SDXL model and variants obtained by incorporating other bias mitigation techniques. The resulting models were compared in terms of fairness metrics, as well as other aspects of generation quality, such as prompt-image alignment. Overall, results showed that PopAlign was able to improve fairness of generated images while maintaining quality with respect to the other evaluated dimensions.

**Strengths:**

- S1: The work proposes a solution to a critical and open research problem.
- S2: Although previous work has shown that there is an inherent trade-off between diversity and quality [1], PopAlign was shown to improve fairness while not greatly harming other quality aspects.
- S3: The authors compared a PopAlign with a diverse set of techniques to mitigate biases in T2I models.
- S4: Beyond autoeval metrics, the authors also evaluated the proposed approach via human evaluation.


[1] Astolfi et al., Consistency-diversity-realism Pareto fronts of conditional image generative models, 2024

**Weaknesses:**

- W1: Unclear generalization of findings to other text-to-image models. The authors only performed experiments with a single base model, namely SDXL, which makes it impossible to assess to what extent the efficacy of PopAlign would transfer to other models.

- W2: Choice of evaluation metrics. CLIP has been extensively shown to not correlate well with human perception of text-to-image alignment [1], making it a poor metric to evaluate prompt-image alignment. Authors could consider, for example, VQA-based metrics such as Gecko and VQAScore [1, 2].

- W3: Lack of statistical significance in the presented results. Results presented in Tables 1, 2, and 3 do not show any notion of dispersion in the reported values, nor statistical significance evaluation. Without this information it is impossible to perform a rigorous comparison of the evaluated methods, making it unclear whether PopAlign is actually being effective in its goal of mitigating gender and ethnicity biases.

- W4: Soundness of human evaluation. In the main text there is no information about how the human evaluation was conducted. Details such as the number of annotators, sample size, annotation template, quality metrics such as inter-annotator agreement are missing. Without these details it is not possible to judge how solid the results are.

- W5: Conflating race and ethnicity. Throughout the text, the authors seem to use the terms race and ethnicity interchangeably, but there are several references pointing out to differences between both identity dimensions, e.g. [3, 4].

[1] Wiles, Olivia, et al. "Revisiting Text-to-Image Evaluation with Gecko: On Metrics, Prompts, and Human Ratings." arXiv preprint arXiv:2404.16820 (2024).

[2] Lin, Zhiqiu, et al. "Evaluating text-to-visual generation with image-to-text generation." European Conference on Computer Vision. Springer, Cham, 2025.

[3] https://en.wikipedia.org/wiki/Ethnicity

[4] McKenzie, Kwame J., and Natasha S. Crowcroft. Race, ethnicity, culture, and science, 1994.

**Questions:**

- Q1: It is unclear to me how the CLIP metric is being computed. CLIP score values fall in the [-1, 1] interval, so it is not clear to me what the CLIP metric on Tables 1, 2, 3, which is greater than 1, is accounting for.

- Q2: Please add details about how the results in Tables 1, 2, and 3 are being computed. What exactly is being presented in the tables? Average scores across all prompts? Also, include confidence intervals and perform statistical significance to allow for proper interpretation of the presented results.

- Q3: Please refer to weakness W4 for missing details on the human evaluation.

- Minor but also relevant: the work has several typos and many words/characters are missing the space between them and the next word/character. For example:
  - Line 101: Space between populations. and Fig. 2.
  - Line 190: noises -> noise.
  - Equations also need punctuation, e.g. commas in Eqs. 2 and 4.

---

> ### Author Response · Authors · 2024-11-21
>
> **W1: Generalization of Findings (Only SDXL Experiments)**
>
> We have added results for SDv1.5 in Table 1, demonstrating PopAlign’s ability to mitigate biases across architectures. Synthetic experiments on Gaussian distributions (Section 4, Appendix H) further validate the method’s general applicability.
>
> **W2: Choice of Evaluation Metrics (CLIP)**
>
> While the limitations of the CLIP metric are recognized, it remains widely used in related works ([1], [2]). We want to point out that we also incorporated additional metrics such as Aesthetic score and HPS in the original paper, where HPS also takes prompt-image alignment into consideration. To strengthen our evaluation, we also report VQA scores for diverse prompts from the LAION-Aesthetic and Personal Descriptors datasets used by aDFT [1]. We find that VQA and CLIP score reflects the same ranking, with our model being equivalent to or slightly worse than the base SDXL, while aDFT[1] performs worse on both metrics.
>
> **W3: Statistical Significance**
>
> We have added statistical significance measurements (e.g., confidence intervals) to Tables 1, 2, and 3 in the revised manuscript. Note that the original Table 2 is now in the Appendix. We also add such metrics for all new experiment tables.
>
> **W4: Soundness of Human Evaluation**
>
> We provide detailed information about the human evaluation process in Section 6.3 (L418) and Appendix E.
> Instructions given to annotators are outlined in Appendix E.
>
> 300 responses were collected for 100 prompts, with 5 images per prompt per model presented to annotators.
> Inter-annotator agreement, measured using Krippendorff’s Alpha, was 0.81, indicating high reliability.
>
> **W5: Conflating Race and Ethnicity**
>
> We apologize for any confusion caused by our terminology. We have clarified that PopAlign focuses exclusively on physical appearances associated with race, not ethnicity, which involves broader sociocultural aspects. The manuscript has been revised.
>
> **Q1: How is the CLIP metric being computed**
>
> We use the official implementation of OpenAI. Specifically, “The values are cosine similarities between the corresponding image and text features, times 100.” according to the official README.
>
> **Q2: What is being computed in table 1,2,3. Please include confidence intervals.**
>
> Yes, you are correct that the average score is being computed. As requested, we incorporate results for all major experiments in the revised pdf.
>
> **Q3:Please refer to weakness W4 for missing details on the human evaluation.**
>
> See responses to W4 as above
>
> **Q4:Minor presentation issues.**
>
> We have fixed typos, missing spaces, punctuations, and other issues. We will further revise the paper for camera-ready if we discover additional issues.
>
>
> [1] Shen, Xudong, et al. "Finetuning text-to-image diffusion models for fairness." arXiv preprint arXiv:2311.07604 (2023).
>
> [2] Wallace, Bram, et al. "Diffusion model alignment using direct preference optimization." Proceedings of the IEEE/CVF Conference on Computer Vision and Pattern Recognition. 2024.
>
> [3] OpenAI, “openai/CLIP”, https://github.com/openai/CLIP, GitHub, 2024

---

> > ### Comment · Reviewer_d8f1 · 2024-11-25
> >
> > Dear authors,
> >
> > Thank you for your response.
> >
> > I appreciate the inclusion of SD 1.5 experiments and confidence intervals in the results, but it is still not clear whether PopAlign indeed offers gains in comparison to the baselines as in many cases the confidence intervals overlap (e.g. discrepancy results in Table 1 for both gender and race).
> > Also, it seems to me the authors are conflating sexuality with sexual orientation, I strongly suggest the authors to clarify this in future versions of the paper.
> >
> > All in all, in light of the clarifications in the rebuttal and the modifications in the experimental section, I raised my score from 3 to 5 but I still believe this work needs better experimental validation prior to be considered for publication.

---

> ### Author Response · Authors · 2024-11-26
>
> We thank the reviewer for raising the score. For additional concerns
>
> **Q:conflating sexuality with sexual orientation**
>
> We have fixed the confusion of sexuality and sexual orientation in the revised version.
>
> **Q:it is still not clear whether PopAlign indeed offers gains in comparison to the baselines as in many cases the confidence intervals overlap**
>
> 1. We believe Reviewer df81 is **conflating the concept of "overlapping confidence interval" with "statistical significance".**  It is entirely possible to have an overlapping 95% confidence interval, while still have the results be statistically significant (p<0.05).  [1][2]. The notion "If two confidence intervals overlap, the difference between two estimates or studies is not significant" is **categorically wrong** and is listed as the "common misinterpretation No. 21" by [2].
>
> 2. **Previous works such as aDFT[3] (ICLR2024 Oral) also reports results with "overlapping confidence intervals"** (their table 1,2,3). In terms of scientific rigor, Popalign offers more rigorous examinations of methods by sampling more images and using more prompts (100 per dataset vs 19 per dataset).  This is reflected in the 95%CI, which is narrower in PopAlign than in aDFT in absolute terms. We respectively suggest the reviewers and AC carefully calibrate the academic standards to be consistent with previous iterations of the conference.
>
> 3. Lastly, we provide the P-value of most major experiments in the below table. Most of the results has (p<0.05). In the small set of cases where P>0.05, it is at least a statistical tie. These results suggest that PopAlign offers an "average improvement" over a diverse set of tasks compared to previous work such as aDFT. We would also like to highlight that Popalign uses considerable less compute than aDFT (Appendix B2)
>
>
>
> | Comparison                        | PopAlign       | 2nd Place     | P-Value   |
> |-----------------------------------|-----------------|----------------|-------|
> |SDXL||||
> | Occupation-G. (Tab. 1)           | 0.18 ± 0.04     | 0.25 ± 0.04    | **0.01**  |
> | Occupation-R. (Tab. 1)           | 0.26 ± 0.05     | 0.31 ± 0.06    | 0.20  |
> | LAION-Aesthetic-G. (Tab. 2)      | 0.14 ± 0.05     | 0.27 ± 0.05    | **0.00**  |
> | LAION-Aesthetic-R. (Tab. 2)      | 0.28 ± 0.05     | 0.30 ± 0.05    | 0.60  |
> | Personal Desc. G. (Tab. 2)       | 0.28 ± 0.05     | 0.33 ± 0.05    | 0.16  |
> | Personal Desc. R. (Tab. 2)       | 0.30 ± 0.03     | 0.43 ± 0.04    | **0.00**  |
> | Age. (Tab. 2)                    | 0.17 ± 0.06     | 0.29 ± 0.06    | **0.01**  |
> | Sexual Orientation. (Tab. 2)     | 0.24 ± 0.16     | 0.62 ± 0.09    | **0.00**  |
> |SDV1.5||||
> | Occupation-G. (Tab. 1)           | 0.15 ± 0.06     | 0.27 ± 0.07    | **0.01**  |
> | Occupation-R. (Tab. 1)           | 0.29 ± 0.05     | 0.36 ± 0.06    | **0.01**  |
>
> G. Gender
> R. Race;
> Lower is better
>
> [1]Austin, Peter C., and Janet E. Hux. "A brief note on overlapping confidence intervals." Journal of vascular surgery 36.1 (2002): 194-195.
>
> [2] Greenland, Sander, et al. "Statistical tests, P values, confidence intervals, and power: a guide to misinterpretations." European journal of epidemiology 31.4 (2016): 337-350.
>
> [3] Shen, Xudong, et al. "Finetuning text-to-image diffusion models for fairness." ICLR 2024.
>
> *miscellaneous: we also corrected a minor mistake in table 2, col 2. ".28±.21" should to be ".28±.05".

---

### Official Review · Reviewer_WFiF · 2024-11-05

**Soundness:** 3
**Presentation:** 2
**Contribution:** 2
**Rating:** 3
**Confidence:** 4

**Summary:**

This paper addresses the issue of inherent biases in generative models, which often stem from biased large-scale training data. The authors propose a method to reduce these biases through population-level preference optimization. Specifically, they generate multiple win-lose pairs for each prompt, which are then used for fine-tuning via preference optimization. Empirical results suggest that this method can help mitigate gender and race biases across a set of 100 prompts.

**Strengths:**

1. The paper is straightforward and easy to follow.

**Weaknesses:**

1. **Lengthy Background and Related Work Sections:** These sections are overly detailed, detracting from the main methodology and potentially causing readers to lose focus on the core contributions of the paper.
2. **Limited Novelty:** The proposed method is relatively straightforward, primarily offering a way to create a fairness-focused pairwise dataset for preference learning. The training technique resembles the approach introduced in Diffusion-DPO, raising concerns about the extent of novelty. Much of the performance gain may be attributed to the newly generated dataset rather than the proposed population-level alignment, which essentially involves using multiple sample sets for each prompt.
3. **Lack of Transparency in Evaluation:** Details about the evaluation process are sparse, with only a mention of 300 prompts used for training and 100 for evaluation. A more transparent evaluation approach would enhance the paper’s credibility.
4. **Narrow Focus on Fairness Issues:** The paper only addresses gender and race biases, which is a limited scope. Expanding to other categories of bias could significantly strengthen the impact and generalizability of the proposed method.

**Questions:**

See weaknesses.

---

> ### Author Response · Authors · 2024-11-21
>
> **Q1: Lengthy Background and Related Work Sections**
>
> We appreciate the reviewer’s constructive feedback. In response, we have removed some sections from the background (e.g., naive RLHF) to streamline the content. Additionally, we have relocated the Related Works section to appear before the Conclusion, improving the paper’s flow. However, we retain key details, such as the DPO formulation in Eq. (1), as they are essential for motivating and deriving our proposed formulation in Eqs. (5-9).
>
> **Q2: Limited Novelty**
>
> We believe there may be some misunderstanding regarding our method. Our primary contribution is not the creation of a fairness-focused pairwise dataset but the development of a **novel training objective for population alignment**. As stated in the Introduction, pairwise preference cannot be defined in our dataset because comparing entities like “a female doctor” and “a male doctor” requires contextualizing within the generation population with many samples, not in isolation using a pair of samples.
>
> Regarding the concern that “training resembles Diffusion-DPO,” this is factually incorrect. Our objective (Eq. 9) involves individual samples, whereas Diffusion-DPO necessitates loading both winning and losing pairs onto each GPU, which is computationally infeasible on our dataset. Even on A100 GPUs, Diffusion-DPO[1] operates with a per-GPU batch size of just 2 images (1 pair), and has to leverage gradient accumulation to handle larger batch sizes. Applying DPO naively to our setup (Eq. 8) cannot even run due to the GPU memory requirements of loading both populations. It will necessitate loading 10 images (1 pair = 5 winning samples and 5 losing samples in our setup). PopAlign provides a novel solution, clearly distinguishing it from Diffusion-DPO.
>
> To further clarify, **PopAlign is the first work that formulates fairness in T2I generation as an alignment problem, a significant conceptual advancement**. Compared to previous works like aDFT, PopAlign achieves stronger performance while requiring significantly fewer resources (Appendix B2).
>
> **We would welcome it if the reviewer can propose some actionable experiments that are feasible** and can show  “the performance gain comes from the proposed population-level alignment objective”, as we have clearly stated that it is not possible to run vanilla DPO on our dataset.
>
> **Q3: Lack of Transparency in Evaluation**
>
> We have enhanced transparency in our evaluation. Specifically:
> 1. Section 5.3 now includes additional evaluation details.
> 2. The full list of prompts used for evaluation is provided in Appendix L.
>
> **Q4: Narrow Focus on Fairness Issues**
>
> We would like to highlight that gender and race are very important factors for bias mitigation. There have been many influential papers written tackling either or both of these issues alone. We acknowledge the importance of expanding the scope of biases addressed. In response:
> 1. We have incorporated additional experiments to address categories like **age and sexuality** (Table 2 in main text, Figure 10 in appendix), demonstrating PopAlign’s generalizability.
> 2. PopAlign is, to the best of our knowledge, the first to mitigate **sexuality biases** in multi-person scenarios, such as “A couple enjoying a picnic.” Addressing such complex setups represents a significant step forward in fairness research.
>
> [1] Wallace, Bram, et al. "Diffusion model alignment using direct preference optimization." Proceedings of the IEEE/CVF Conference on Computer Vision and Pattern Recognition. 2024.

---

> > ### Author Response · Authors · 2024-12-01
> >
> > Dear Reviewer WFiF
> >
> > As we are approaching the end of discussion period, we would greatly appreciate your feedbacks on our rebuttals. We have provided additional experiments and discussions as requested, such as expanding to other bias categories and improving the transparency of evaluations by providing the full prompts.
> >
> > Best, Authors.

---

> > > ### Author Response · Authors · 2024-12-02
> > >
> > > Dear Reviewer WFiF
> > >
> > > As we are approaching the **last day** of discussion period, we would greatly appreciate your feedbacks on our rebuttals. We have provided additional experiments and discussions as requested, such as expanding to other bias categories and improving the transparency of evaluations by providing the full prompts.
> > >
> > > Best, Authors.

---

> > > > ### Author Response · Authors · 2024-12-03
> > > >
> > > > Dear Reviewer WFiF
> > > >
> > > > Today is the **last day** of discussion period, we would greatly appreciate your feedbacks on our rebuttals. We have provided additional experiments and discussions as requested, such as expanding to other bias categories and improving the transparency of evaluations by providing the full prompts.
> > > >
> > > > Best, Authors.

---

### Author Response · Authors · 2024-11-21

We have revised our paper according to constructive feedbacks. The main changes are
1. **Moved appendix to the same pdf as the main text**
2. We cut down background section, and moved related works to before conclusion for better flow of the paper ( WFiF)
3. We include a full list of prompts in appendix L ( WFiF,d8f1,d3aE)
4. We add new experiments for age and sexuality debiasing in Table 2. We also add experiments on additional diverse prompts from LAION-Aesthetics and Personal Descriptors. (WFiF,d8f1,d3aE)
4. We incorporated results for SDv1.5 in Table1 (d8f1)
5. We added confidence intervals for all main experiments  (d8f1)
6. We provided human evaluation details in Section 6.3. The full prompt is provided in Appendix E (d8f1,d3aE)
7. We changed the term ethnicity to race (d8f1)
8. We provided additional discussion on computation cost in Appendix B2 (WFiF, d8f1,d3aE)

---

### Note · Authors · 2025-01-22

I have read and agree with the venue's withdrawal policy on behalf of myself and my co-authors.